# The genetic architecture of the load linked to dominant and recessive self-incompatibility alleles in *Arabidopsis halleri* and *Arabidopsis lyrata*

**Audrey Le Veve†, Mathieu Genete, Christelle Lepers-Blassiau, Chloé Ponitzki, Céline Poux, Xavier Vekemans, Eleonore Durand, Vincent Castric***

Univ. Lille, CNRS, UMR 8198 – Evo-Eco-Paleo, Lille, France

**\*For correspondence:**
Vincent.Castric@univ-lille.fr

**Present address:** †Department of Botany, Faculty of Science, Charles University, Prague, Czechia

**Abstract** The long-term balancing selection acting on mating types or sex-determining genes is expected to lead to the accumulation of deleterious mutations in the tightly linked chromosomal segments that are locally 'sheltered' from purifying selection. However, the factors determining the extent of this accumulation are poorly understood. Here, we took advantage of variations in the intensity of balancing selection along a dominance hierarchy formed by alleles at the sporophytic self-incompatibility system of the Brassicaceae to compare the pace at which linked deleterious mutations accumulate among them. We first experimentally measured the phenotypic manifestation of the linked load at three different levels of the dominance hierarchy. We then sequenced and phased polymorphisms in the chromosomal regions linked to 126 distinct copies of *S*-alleles in two populations of *Arabidopsis halleri* and three populations of *Arabidopsis lyrata*. We find that linkage to the *S*-locus locally distorts phylogenies over about 10–30 kb along the chromosome. The more intense balancing selection on dominant *S*-alleles results in greater fixation of linked deleterious mutations, while recessive *S*-alleles accumulate more linked deleterious mutations that are segregating. Hence, the structure rather than the overall magnitude of the linked genetic load differs between dominant and recessive *S*-alleles. Our results have consequences for the long-term evolution of new *S*-alleles, the evolution of dominance modifiers between them, and raise the question of why the non-recombining regions of some sex and mating type chromosomes expand over evolutionary times while others, such as the *S*-locus of the Brassicaceae, remain restricted to small chromosomal regions.

## eLife assessment

This study presents **valuable** empirical work and simulations that are relevant for the evolution of genetic load linked to self-incompatibility alleles in two *Arabidopsis* species. The evidence supporting the findings is **solid**, although it remains to be seen how generalisable the conclusions are beyond the specific system investigated here, not least because the statistical significance varied between the two species. The work will be of relevance to geneticists interested in the evolution of allelic diversity in similar systems.

## Introduction

The existence of sexes or mating types leads to one of the strongest forms of long-term balancing selection and is often associated with clusters of polymorphisms around sex/mating-type-controlling regions kept together by structural rearrangements. In some cases, such rearrangements can span

almost entire chromosomes, for example, sex chromosomes in mammals (*Katsura et al., 2012*) or mating-type chromosomes in ascomycete fungi (*Hartmann et al., 2021*), while in others they remain limited to relatively small genomic regions, for example, chromosomal inversions controlling male reproductive morphs in the ruff (*Lamichhaney et al., 2016*), mating-type loci in some basidiomycete fungi, segregating indels controlling pin vs. thrum floral morphs in *Primula* (*Cocker et al., 2018*). The long-term balancing selection acting on these systems is expected to lead to the accumulation of deleterious mutations in the tightly linked chromosomal segments that are 'sheltered' from purifying selection by the presence of the balanced polymorphism (*Uyenoyama, 1997*; *Uyenoyama, 2005*). These deleterious mutations can have drastic short- and long-term consequences for the evolution of the species, and determining the processes by which they accumulate is crucial to understand how the rearranged regions can either expand along the chromosomes or conversely remain restricted to limited genomic tracts (*Jay et al., 2021*; *Jay et al., 2022*).

Self-incompatibility (SI) is a genetic mechanism allowing recognition and rejection of self-pollen by hermaphrodite individuals, thereby preventing inbreeding and promoting outcrossing in hermaphroditic plant species (*Nettancourt, 2001*). In the Brassicaceae family, SI is controlled by a single non-recombining chromosomal region, the *S*-locus (*Schopfer et al., 1999*; *Kusaba et al., 2001*). SI is one of the most prominent examples of long-term balancing selection (*Uyenoyama, 2003*), and as such deleterious mutations are expected to accumulate in very close genetic linkage to the *S*-alleles because of the indirect effects of linked selection (*Uyenoyama, 2005*). Population genetics models predict that recessive deleterious variants should accumulate within specific *S*-allele lineages (*Uyenoyama, 2003*; *Llaurens et al., 2009b*) and should then be reshuffled among them by recombination. However, due to the technical difficulty of phasing polymorphisms, this process has rarely been characterised in detail (*Castric and Vekemans, 2004*).

A key feature of sporophytic SI systems, also shared by sex chromosomes, is the existence of dominance interactions between *S*-alleles. While most individuals are heterozygous at the *S*-locus and thus carry two different *S*-alleles, only one of them is generally expressed at the phenotypic level. This is especially true for the pollen specificity, where *S*-alleles follow a complex genetic dominance hierarchy (*Llaurens et al., 2008*; *Durand et al., 2014*). In *Arabidopsis lyrata* and *Arabidopsis halleri*, the S-alleles are distributed into four classes of dominance, with increasing dominance from class I to class IV and linear dominance also observed inside some of the classes. Similar patterns of dominance between S-alleles occur in *Brassica* species, but with only two classes (*Hatakeyama et al., 1998*; *Kakizaki et al., 2003*; *Yasuda et al., 2017*). This system is genetically determined and controlled by small RNAs molecules produced by dominant *S*-alleles that are able to target and repress expression of the recessive *S*-alleles in pollen (*Durand et al., 2014*; *Yasuda et al., 2017*). The evolutionary properties of *S*-alleles are expected to vary in a predictable manner along the dominance hierarchy because balancing selection acts more strongly on dominant than on recessive *S*-alleles as the latter are often masked at the phenotypic level (*Billiard et al., 2007*). As a result, the dynamics of accumulation of deleterious variation may differ in close linkage with dominant vs. recessive *S*-alleles. Specifically, recessive *S*-alleles can form homozygous combinations in natural populations more often than dominant *S*-alleles (*Billiard et al., 2007*), such that recombination can occur occasionally between distinct gene copies of the same recessive *S*-allele, providing the opportunity for the linked recessive deleterious mutations to be purged from within the *S*-locus itself. In addition, because recessive *S*-alleles reach higher population frequencies (*Billiard et al., 2007*; *Llaurens et al., 2008*), purifying selection on linked deleterious variants is expected to have higher efficacy among gene copies of recessive than dominant *S*-alleles. This is expected to result in a higher fixation probability of deleterious variants linked to the class of dominant *S*-alleles than to the class of recessive *S*-alleles (*Llaurens et al., 2009b*). Thus, the level of dominance of S-alleles determines the intensity of purifying selection acting upon them. This situation closely resembles that of the differential evolution of sex chromosomes, where Y chromosomes (similar to dominant S-alleles) tend to accumulate more deleterious variation than X chromosomes (similar to recessive S-alleles; *Llaurens et al., 2009b*; *Goubet et al., 2012*). Empirical support for this simple prediction has been conflicting, though. Based on phenotypic measurements in *A. halleri*, *Llaurens et al., 2009b* observed a decrease of fitness associated by enforced homozygosity for one of the most dominant *S*-alleles (Ah15) but not for the most recessive *S*-allele of the allelic series (Ah01). In contrast, *Stift et al., 2013* observed no effect of dominance on the genetic load linked to three dominant vs. recessive *S*-alleles in a natural population of the closely related *A.*

*lyrata*. Hence, the data available so far are inconclusive, but are restricted to very small numbers of *S*-alleles. They are also based on inherently limited phenotypic measurements, seriously limiting the power of the comparisons, and preventing proper generalisation of the effect of the intensity of balancing selection on the accumulation of linked deleterious variation.

In this study, we combined phenotypic, genomic, and theoretical approaches to finely dissect the patterns of accumulation of deleterious variation linked to the *S*-locus supergene in *A. halleri* and *A. lyrata*, depending on dominance levels of S-allele. We first extended the phenotypic approach of *Llaurens et al., 2009b* to a series of additional *S*-alleles from the same local *A. halleri* population to evaluate the effect of *S*-allele dominance on the sheltered load. We then used parent–offspring trios and targeted genome re-sequencing to directly quantify the accumulation of putative deleterious mutations linked to phased dominant vs. recessive *S*-alleles in two *A. halleri* and three *A. lyrata* natural populations. Finally, we used stochastic simulations to refine the theoretical predictions about the patterns of accumulation of recessive deleterious mutations linked to dominant vs. recessive *S*-alleles. Overall, our results provide a more nuanced view of the effect of the intensity of balancing selection on the sheltered load, in which the structure of the sheltered load rather than its magnitude differs among *S*-alleles from different dominance classes.

## Results
### The genetic load linked to the *S*-locus varies among *S*-alleles, but is not correlated with dominance

We first expanded the experimental approach of *Llaurens et al., 2009b* to phenotypically evaluate the effect of *S*-allele dominance on the intensity of the sheltered load. The previous study focused on three *S*-alleles (Ah01, Ah02, and Ah15; *Llaurens et al., 2009b*). Here we included two *S*-alleles from the same local population (Nivelle, France): Ah03 and Ah04, and included Ah01 again for comparative purposes. In the *Arabidopsis* genus, *S*-alleles have been shown to form a complex dominance hierarchy (*Llaurens et al., 2008*; *Durand et al., 2014*). This hierarchy is largely associated with the phylogeny of *S*-alleles (*Durand et al., 2014*), and at least four phylogenetic classes (I, II, III, and IV) have been described, from the most recessive (class I) to the most dominant of *S*-alleles (class IV). Dominance interactions also exist among *S*-alleles within classes, such that these five *S*-alleles form the following dominance hierarchy (*Llaurens et al., 2008*; *Durand et al., 2014*): Ah01<Ah03<Ah02<Ah04<Ah15, from the most recessive (Ah01) to the most dominant (Ah15). To reveal the linked load, we enforced homozygosity at the *S*-locus using controlled crosses between parental individuals sharing a given *S*-allele that was masked by different dominant *S*-alleles (e.g. to obtain $Ah_xAh_x$ homozygotes we deposited pollen from a $Ah_xAh_y$ plant, where $Ah_y > Ah_x$, on pistils of a $Ah_xAh_z$ plant where z≠y, or on a $Ah_xAh_x$ pistil when available; *Supplementary file 1a*). We obtained 399 offspring from a total of six

**Table 1.** Proportion of *S*-locus homozygous offspring having reached the reproductive stage for three different *S*-alleles.

The test is performed relative to the expected proportion of homozygous genotypes in the offspring (25% when both parents are heterozygous; 50% when one of the parents is homozygous and the other heterozygous).

| *S*-allele | Level of dominance | Number of seedlings having reached the reproductive stage | Observed proportion of homozygotes | Ratio of the observed to expected proportion of homozygous genotypes (p-value*) | Number of heterozygotes with the *S*-allele (p-values*) |
|---|---|---|---|---|---|
| Ah01 | I | 35 | 0.29 | 1.14 (0.76) | 19 (0.40; 0.90) |
| Ah03 | II | 27 | **0.074** | **0.3 (0.02)** | 17 (1; 0.69) |
| Ah04 | III | 96 | 0.479 | 0.96 (0.39) | 50 |

Values departing from Mendelian expectations are figured in bold. For Ah04, the maternal parent was Ah04/Ah04, so all heterozygous offspring carried the S-allele.

*p-Values were obtained after 10,000 random permutations, respectively.

such crosses. Note that our experimental procedure differs slightly from that of *Llaurens et al., 2009b* in that their procedure required a $CO_2$ treatment to bypass the SI system and obtain selfed offspring, while here we took advantage of the dominance interactions to obtain outcrossed *S*-locus homozygous individuals that we phenotypically compared to their full-sibs with *S*-locus heterozygous genotypes. Note also that the *S*-locus homozygous offspring we obtained contain distinct gene copies of a given *S*-allele lineage. Hence, they could in principle carry distinct suites of linked deleterious mutations in case these mutations segregate within *S*-allele lineages.

We first tested whether homozygosity at the *S*-locus affected survival by measuring for each cross the proportion of homozygotes at the *S*-locus reaching the reproductive stage for three *S*-alleles (in two replicate families per *S*-allele; *Supplementary file 1a*). The proportion of Ah01/Ah01 and Ah04/Ah04 homozygotes surviving to the reproductive stage was consistent with Mendelian expectations in their respective families. However, we observed a significant decrease of Ah03/Ah03 homozygotes at the reproductive stage compared with Mendelian expectations (*Supplementary file 1a*), whereas the observed proportion of the Ah03 *S*-allele among heterozygous individuals did not depart from

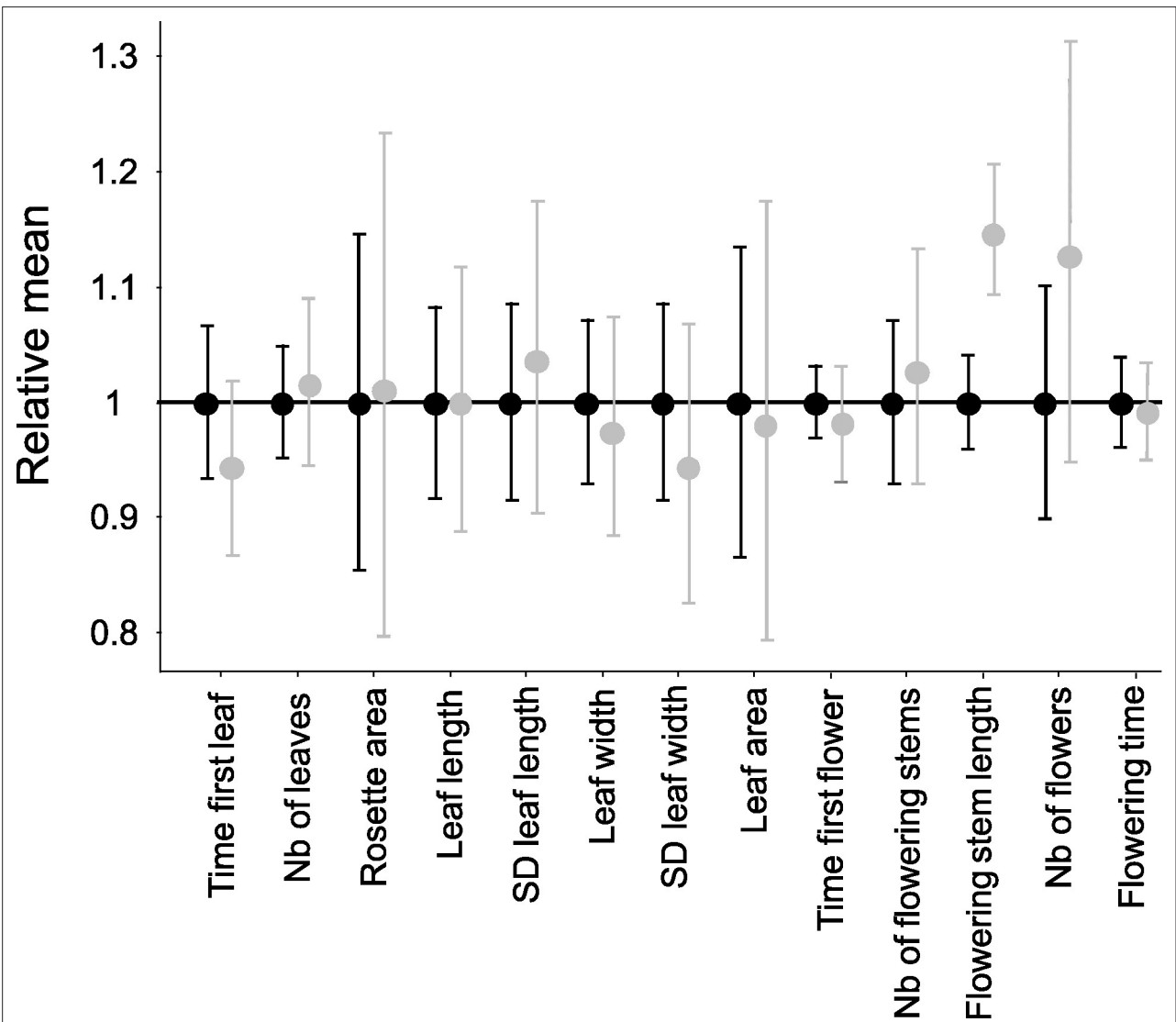

**Figure 1.** Effect of homozygosity at the *S*-locus on 13 phenotypic traits compared to heterozygotes. For each trait, the phenotypic values in homozygotes (in grey; n=72) were normalised relative to the mean phenotypic values in heterozygotes (in black; n=86). The point represent the mean and the barres represent the standard deviations. The differences of distributions were tested by 10,000 random permutations.

The online version of this article includes the following figure supplement(s) for figure 1:

**Figure supplement 1.** Experimental protocol.

expectations (2/3 = 0.67; *Table 1*). Thus, the increased mortality is associated with Ah03 homozygosity, rather than with a lower performance of individuals carrying the Ah03 *S*-allele itself. Overall, a genetic load was thus observed linked to the Ah03 *S*-alleles, which is at an intermediate level of dominance, but neither to the most dominant (Ah04) nor to the most recessive (Ah01) *S*-allele. Hence, these observations do not support a positive correlation between *S*-allele dominance and the magnitude of the sheltered load.

Next we measured 13 vegetative and reproductive traits in the resulting families and compared offspring that were homozygous for their *S*-alleles with their full sibs that were heterozygous (*Figure 1—figure supplement 1*). We first used permutations to test whether the mean trait value of homozygotes differed from that in heterozygotes. Overall, with a single exception, we found no effect of homozygosity at the *S*-locus on variation of the traits measured (*Figure 1*; *Supplementary file 1b*). The maximum length of flowering stems was the exception to this general pattern, with longer reproductive stems for *S*-locus homozygous than heterozygous genotypes, hence in the opposite direction from our expectation of lower fitness in homozygotes. For this trait, there was significant variation among replicate families for homozygotes of the recessive allele Ah01 but not of the dominant allele Ah04 (*Supplementary file 1c*). We then used generalised linear models (GLM) to evaluate the effect of dominance (considered as a continuous variable with fixed effect) on the mean phenotypic value of homozygotes compared to heterozygotes for each trait (*Supplementary file 1d*; treating family of origin, attacks by phytopathogens, phytophagous and oxidative stress as random effects whenever necessary). We also observed no effect of *S*-allele dominance on the contrast between *S*-locus homozygotes and heterozygotes for any of these traits. A single of the 13 traits was an exception to this general pattern, but again the effect was in the opposite direction from our expectation, with an earlier rather than delayed appearance of the first leaf for homozygotes of more dominant *S*-alleles (*Supplementary file 1d*). Overall, our phenotypic results confirmed the presence of a detectable linked load on some phenotypic traits (survival; time to produce the first leaf), but we could not replicate the observation of *Llaurens et al., 2009b* that dominant *S*-alleles carry a more severe deleterious load than recessive *S*-alleles, even though our samples were obtained from the same local population.

## *S*-alleles are associated with specific sets of tightly linked mutations

The model of the sheltered load assumes that distinct *S*-allele lineages carry specific sets of linked deleterious mutations, but to our knowledge this prediction was never tested directly. We combined a parent–offspring trio approach with sequencing of the *S*-locus flanking regions to phase the mutations segregating in the *S*-locus flanking regions with their respective *S*-alleles. Briefly, we used a previously developed sequence capture protocol specifically targeting the nucleotide sequences over 75 kb on each side of the *S*-locus along with a series of 100 control regions from throughout the genome (*Le Veve et al., 2023*), and analysed nucleotide sequence polymorphism (including only invariant and biallelic SNPs), based on the *A. lyrata* reference genome (*Hu et al., 2011*). We define a haplotype as a unique combination of mutations along the phased chromosome, and an *S*-allele lineage as the collection of gene copies of a given functional *S*-allele (different functional *S*-alleles are distinguished based on their strong sequence divergence at the *S*-locus pollen and pistil genes). Different gene copies within an *S*-allele lineage can thus be associated with distinct linked haplotypes in the flanking regions. The *S*-alleles were identified based on short reads sequences according to a previously published method (*Genete et al., 2020*). We analysed two closely related *A. halleri* populations from Europe (Nivelle and Mortagne) and three allogamous *A. lyrata* populations from North America (IND, PIN, and TSS; *Foxe et al., 2010*). Overall, we were able to reconstruct 34 haplotypes linked to a total of 12 distinct *S*-allele lineages in Nivelle, 38 haplotypes linked to 11 distinct *S*-allele lineages in Mortagne, and 16, 22, and 16 haplotypes associated with 6, 7, and 5 distinct *S*-allele lineages in populations IND, PIN, and TSS, respectively (*Supplementary file 1e*). Nine of the *S*-alleles were shared between the two *A. halleri* populations (Ah01, Ah03, Ah04, Ah05, Ah12, Ah20, Ah24, Ah25, and Ah59). In the populations of *A. lyrata*, four *S*-alleles were shared between PIN and TSS (Ah01*, Ah03*, Ah18*, and Ah63*), five *S*-alleles were shared between PIN and IND (Ah01*, Ah03*, Ah46*, and Ah63*), four *S*-alleles were shared between IND and TSS (Ah01*, Ah03*, Ah31*, and Ah63*), and three were shared across all three (Ah01*, Ah03*, and Ah63*). Note that for convenience, we used *A. halleri* notations (with the addition of a *) to refer to the trans-specifically shared *A. lyrata* *S*-alleles. Altogether, we were able to obtain the phased flanking sequences of 126 *S*-locus haplotypes, comprising a total of

4854 variable sites. This provides considerable power to evaluate the local accumulation of linked mutations across S-alleles of different levels of dominance and examine their patterns of conservation between populations and between species.

Mutations in the S-locus flanking regions can be exchanged between S-alleles by recombination and between local populations by migration (*Charlesworth, 2006*). The relative time scale of these two processes (recombination vs. migration) determines the distribution of the linked mutations. To capture the chromosomal extent of this effect of linkage to S-alleles, we developed a new phylogenetic method comparing the likelihood of two contrasted topologies of interest in overlapping windows along the chromosome: (1) the topology clustering haplotypes by the populations where they came from vs. (2) the topology clustering them by the S-allele to which they are linked (*Figure 2A*). This allowed us to evaluate the progressive shift from a predominant topology by S-alleles close to the S-locus to a topology by populations further along the chromosome and in unlinked control regions (*Figure 2B*). The difference in log likelihood between the two topologies decreased significantly with distance to the S-locus (Pearson coefficient = –0.015 and –0.010 for *A. halleri* and *A. lyrata,* respectively; p-values<$2^{e-16}$). In *A. halleri*, the topology grouping haplotypes by populations became more likely than the topology grouping them by S-alleles at a distance of around 30 kb from the S-locus, but even at a distance of 50 kb the phylogenetic structure was still different from that in regions unlinked to the S-locus used as controls for the genomic background (*Le Veve et al., 2023*; *Figure 2B*). In *A. lyrata,* the shift was even more rapid (within 10–15 kb), although we note that the phylogenetic structure of the control regions was less resolved (*Figure 2B*). To evaluate these patterns more directly, we first examined the data using a major component analysis (MCA, a modified version of PCA adapted to binary data, *Figure 2—figure supplement 1B*, *Figure 2—figure supplement 2B*) and using simple phylogenetic reconstructions (*Figure 2B—figure supplement 3–6*). We confirmed that haplotypes linked to a given S-allele tended to cluster together in the most tightly linked region, and that this grouping by S-alleles was progressively lost in favour of a grouping by population of origin in the most distant regions. Following *Kamau et al., 2007*, we compared the fixation index $F_{ST}$ among local populations and among S-alleles in *A. lyrata* and *A. halleri*. In both *A. halleri* and *A. lyrata*, $F_{ST}$ values among S-alleles were high in regions close to the S-locus and quickly decreased to reach the background level (*Figure 2B—figure supplement 8*) as the distance from the S-locus increased. In parallel, the differentiation among populations followed roughly the opposite pattern, that is, it was initially low in regions close to the S-locus (as expected under strong balancing selection) and increased up to background level within the first few kilobases (*Figure 2—figure supplement 7*). In line with our phylogenetic analysis, differentiation between populations started to exceed differentiation between S-alleles much closer to the S-locus in the *A. lyrata* than in the *A. halleri* populations (*Figure 2—figure supplements 3–5* and *Figure 2—figure supplement 7B*). Finally, we explored the fine-scale patterns of association within populations between individual S-alleles and SNP in the linked and the control regions (*Figure 2—figure supplement 8*). As expected, the vast majority of significant associations were found for the most closely linked SNPs. With a single exception, all S-alleles were associated with unique SNPs in the 50 kb region around the S-locus, albeit with substantial heterogeneity among S-alleles in the patterns and extent of associations that they show (*Figure 2—figure supplement 8*). Overall, our results indicate that due to limited recombination, the S-alleles carry a specific set of polymorphic sites in the linked region. This association fades away for more distant sites over a few kilobases, where population structure becomes predominant, as in the rest of the genome. Hence, different S-alleles are associated with specific sets of tightly linked mutations, but only within 10–30 kb.

## No overall evidence that dominant *S*-alleles accumulate more linked deleterious mutations

*Llaurens et al., 2009b* predicted that recessive deleterious mutations should fix more readily when linked to dominant S-alleles than when linked to recessive S-alleles. To test this prediction, we investigated the correlation between the level of dominance of the S-alleles and their total number of zerofold degenerate mutations ($S_{0f}$) or the ratio of zerofold to fourfold mutations ($S_{0f}/S_{4f}$) for the phased haplotypes, assuming that the vast majority of zerofold degenerate mutations are deleterious. Based on the results presented above and the results of our previous study (*Le Veve et al., 2023*), for the rest of our analyses we focused on the phased haplotypes over the first 25 kb on either side of the

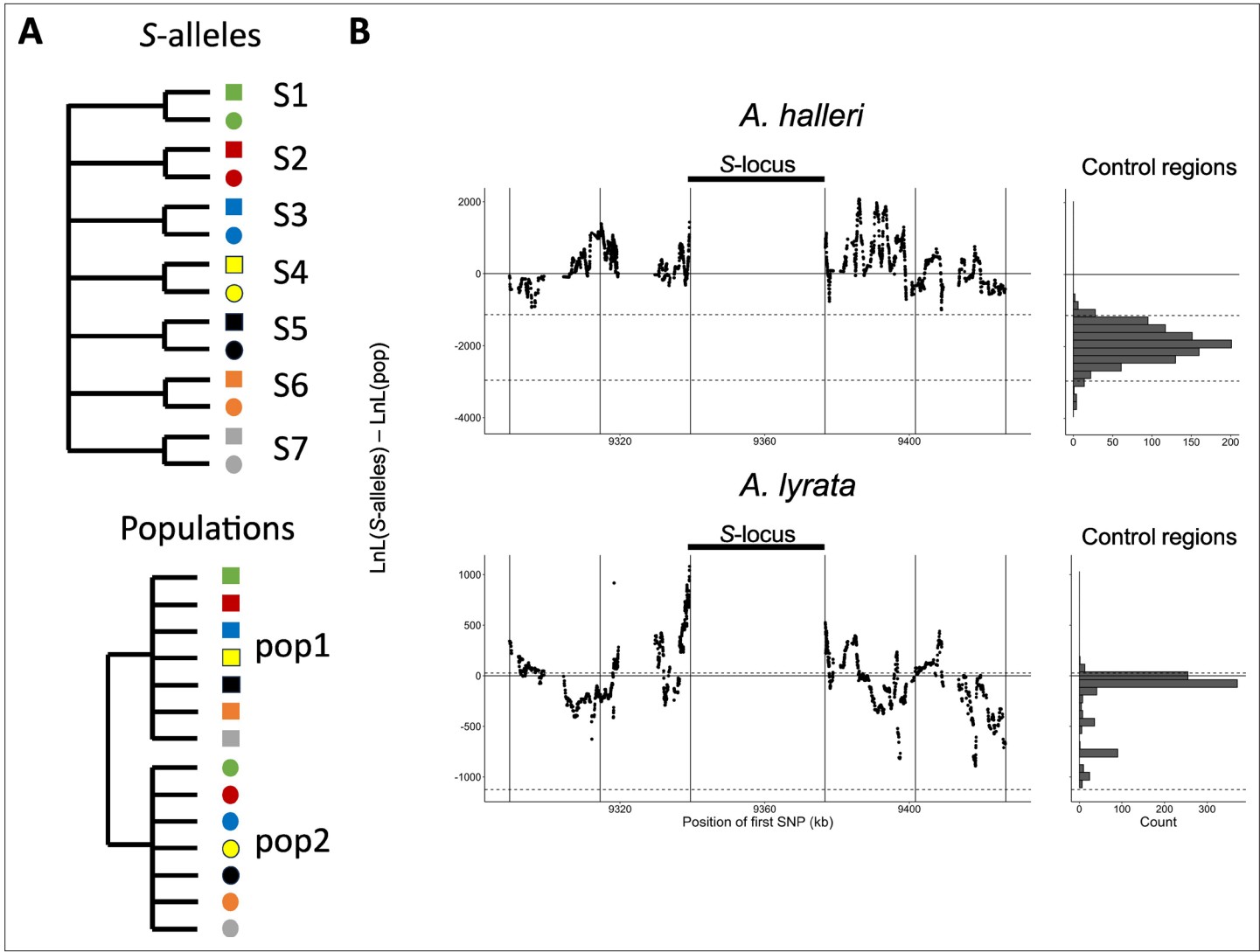

**Figure 2.** Linkage to the *S*-locus locally distorts the phylogenetic relationships. (**A**) The two topologies of interest cluster haplotypes either by the *S*-allele to which they are linked (top) or by the populations where they came (bottom). Different *S*-alleles are represented by symbols of different colours, different populations of origin are represented by symbols of different shapes. (**B**) Difference in log likelihood of the two topologies of interest. Dots correspond to the difference in log likelihood for overlapping series of 50 SNPs around the *S*-locus for *A. halleri* (top panel) and *A. lyrata* (bottom panel). Positive values correspond to chromosomal positions where the topology by *S*-alleles explains the phylogeny of haplotypes better than the topology by populations. The right panels show the difference in log likelihood in the control regions. 2.5 and 97.5 percentiles of the distribution in the control regions are indicated by dashed lines.

The online version of this article includes the following figure supplement(s) for figure 2:

**Figure supplement 1.** Analysis of major components obtained for haplotypes of *A. halleri* of the Nivelle (black dots) and Mortagne (grey dots) populations based on SNPs in the first 5 kb, between 5 and 25 kb, and between 25 kb and 50 kb away from the *S*-locus.

**Figure supplement 2.** Analysis of major components (AMC) obtained for haplotypes of *A. lyrata* (of the PIN: grey dots, IND: red dots; and TSS: blue dots) populations based on the SNPs in the first 5 kb, between 5 and 25 kb, and between 25 kb and 50 kb away from the *S*-locus.

**Figure supplement 3.** Phylogenetic tree obtained by maximum likelihood for haplotypes of *A. halleri* (populations Nivelle and Mortagne) across the first 25 kb flanking the *S*-locus.

**Figure supplement 4.** Phylogenetic tree obtained by maximum likelihood for haplotypes of *A. halleri* (populations Nivelle and Mortagne) based on the nucleotide positions between 25 kb and 50 kb away from the *S*-locus.

**Figure supplement 5.** Phylogenetic tree obtained by maximum likelihood for haplotypes of *A. lyrata* (populations PIN, IND, TSS) across the first 5 kb flanking the *S*-locus.

**Figure supplement 6.** Phylogenetic tree obtained by maximum likelihood for haplotypes of *A. lyrata* (populations PIN, IND, TSS) based on the nucleotide positions between 5 kb and 10 kb away from the *S*-locus.

*Figure 2 continued on next page*

*Figure 2 continued*

**Figure supplement 7.** The genetic structure of SNPs in the *S*-locus flanking regions in *A. halleri* and *A. lyrata*.

**Figure supplement 8.** Patterns of genetic associations between *S*-alleles and SNPs across the genome.

*S*-locus. We found no overall effect of dominance on $S_{0f}$ (p-values = 0.54 and 0.07 for *A. halleri* and *A. lyrata*, respectively; *Figure 3*; *Supplementary file 1f*) or $S_{0f}/S_{4f}$ (p-values = 0.54 and 0.07 for *A. halleri* and *A. lyrata*, respectively; *Supplementary file 1f*). Extending the analysis to all non-synonymous mutations or to deleterious mutations predicted by SIFT4G and by SNPeff led to identical conclusions (*Supplementary file 1f*). Overall, our genomic results did not confirm the prediction that dominant *S*-alleles accumulate a larger number of putatively deleterious mutations in their linked regions. We note that the particular *S*-allele whose sheltered load was quantified in *Llaurens et al., 2009b* (Ah15, red arrow in *Figure 3A*) appears to be one of the *S*-alleles associated with the highest number of zerofold degenerate mutations among all *S*-alleles of the most dominant class (class IV).

## The structure of the linked genetic load differs between dominant and recessive *S*-alleles

Theory predicts that dominant *S*-alleles should fix linked recessive deleterious mutations with a higher probability than recessive *S*-alleles (*Llaurens et al., 2009b*), but in natural populations we observed no difference in the total number of putatively deleterious linked to dominant vs. recessive *S*-alleles. To clarify this discrepancy, we took advantage of our sequencing of multiple copies of *S*-alleles to consider separately the fixed and the segregating mutations linked to each of the *S*-allele lineages. For each population, we included only mutations that were segregating and excluded those that were locally fixed. In agreement with the prediction of *Llaurens et al., 2009b*, we observed that lineages of dominant *S*-alleles do indeed tend to fix deleterious mutations more readily (*Figure 4*; *Supplementary file 1f*). This conclusion held true when extending the analysis to all non-synonymous mutations

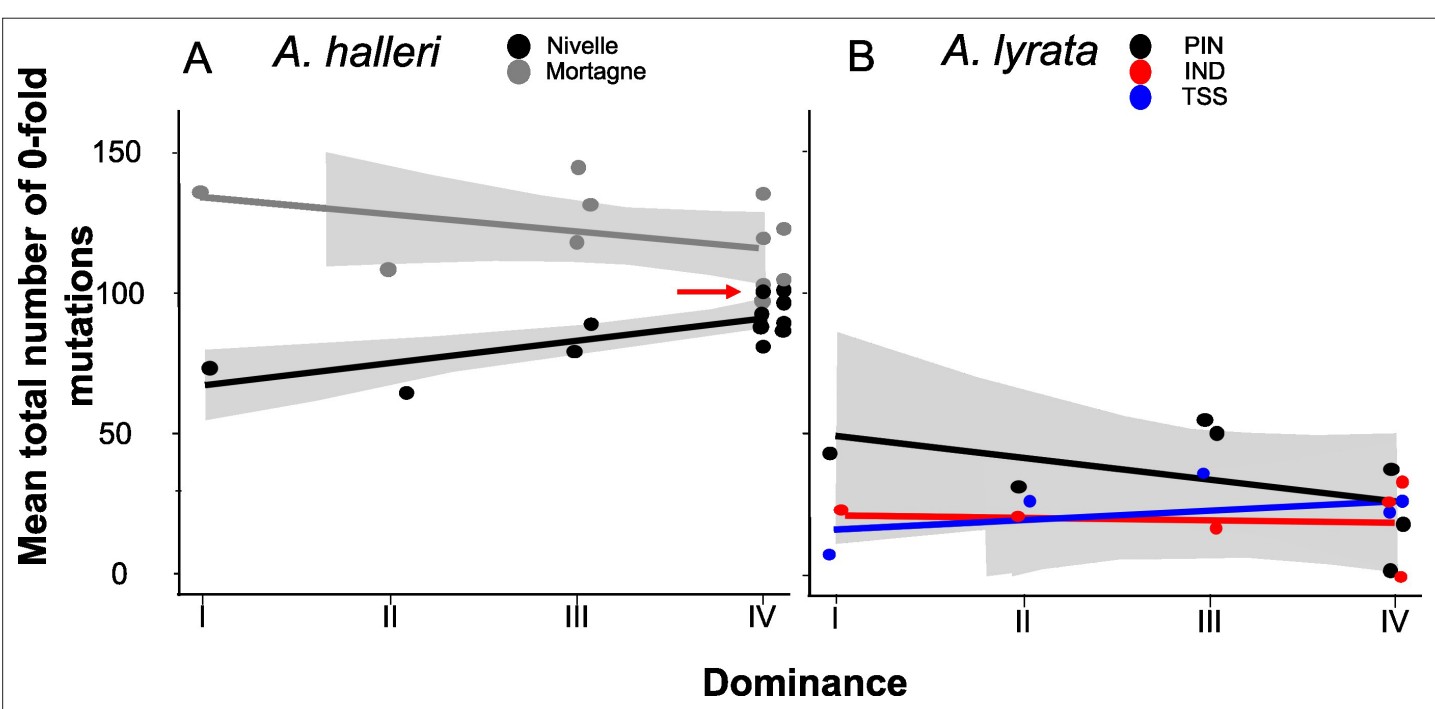

**Figure 3.** No overall effect of *S*-allele dominance on the total number of zerofold degenerate mutations ($S_{0f}$) in the linked genomic regions within 25 kb. Each dot represents the mean number of mutations observed among haplotypes linked to one *S*-allele in one population. The correlations evaluated by a generalised linear model (GLM) are represented by lines, with confidence intervals represented in grey. The dominance was considered a continuous variable. (**A**) *A. halleri*. Black dots correspond to the Nivelle population, grey dots to the Mortagne population. The red arrow points to the copy of Ah15, corresponding to the *S*-allele whose sheltered load was phenotypically characterised by *Llaurens et al., 2009bLlaurens et al., 2009b* in Nivelle. (**B**) *A. lyrata*. Red dots correspond to the IND population, black dots to the PIN population, and blue dots to the TSS population.

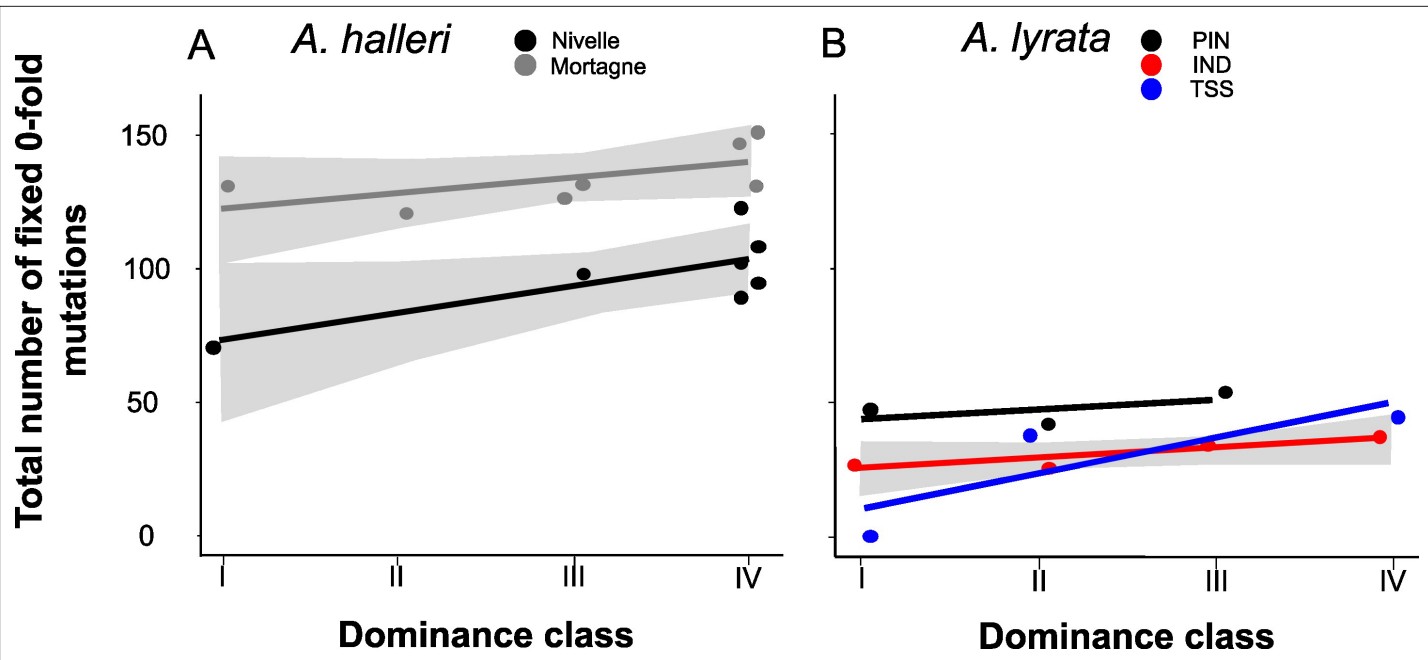

**Figure 4.** The number of zerofold degenerate mutations fixed in the 25 kb regions flanking the *S*-locus increases with dominance of the *S*-allele associated. Each dot represents the value obtained for haplotypes linked to one *S*-allele in one population. The correlations evaluated by a generalised linear model (GLM) are represented by lines, with confidence intervals represented in grey. The dominance was considered as a continuous variable. (**A**) *A. halleri*. Black dots correspond to the Nivelle population, grey dots to the Mortagne population. (**B**) *A. lyrata*. Red dots correspond to the IND population, black dots to the PIN population, and blue dots to the TSS population.

and to the lowly and the moderately deleterious mutations predicted by SNPeff (*Supplementary file 1f*). This was also true using SIFT4G to identify deleterious mutations, with the only exception of a nonsignificant correlation for *A. halleri*, which might be due to the low number of nucleotide sites included in the SIFT4G database, resulting in low power to detect differences (*Supplementary file 1f*). The fact that dominant *S*-alleles tend to fix deleterious mutations more readily but do not accumulate a larger total number of deleterious mutations is explained by the fact that the structure of the genetic load differs between dominant and recessive *S*-alleles: the dominant *S*-alleles tend to have more fixed deleterious mutations, but the recessive *S*-alleles compensate by accumulating a larger number of segregating mutations, resulting in similar numbers of deleterious mutations overall in most of the populations.

Motivated by these empirical observations, we built upon the model by *Llaurens et al., 2009b*, who showed that linked deleterious mutations (especially fully recessive ones) are expected to fix within dominant S-allele lineages more readily than within recessive S-allele lineages. Here, we adapted the model to focus not only on fixed deleterious mutations, but also on those that are segregating within allelic lineages. Our stochastic simulations confirmed that, at equilibrium, dominant *S*-alleles tend to accumulate a larger number of recessive deleterious mutations that are fixed among gene copies within *S*-allele lineages (*Figure 5A*). In contrast, the number of segregating linked mutations was higher for recessive than for dominant *S*-alleles (*Figure 5B*). These two effects eventually compensate each other, such that in the end the mean number of linked deleterious mutations per copy of S-allele was not expected to change between dominant and recessive S-alleles (*Figure 5C*). These predictions are in line with our genomic observations and suggest that the dominance level of *S*-alleles modifies the structure of the genetic load they shelter: dominant *S*-alleles accumulate more fixed deleterious mutations, but recessive *S*-alleles accumulate more segregating mutations, resulting in an equivalent load overall.

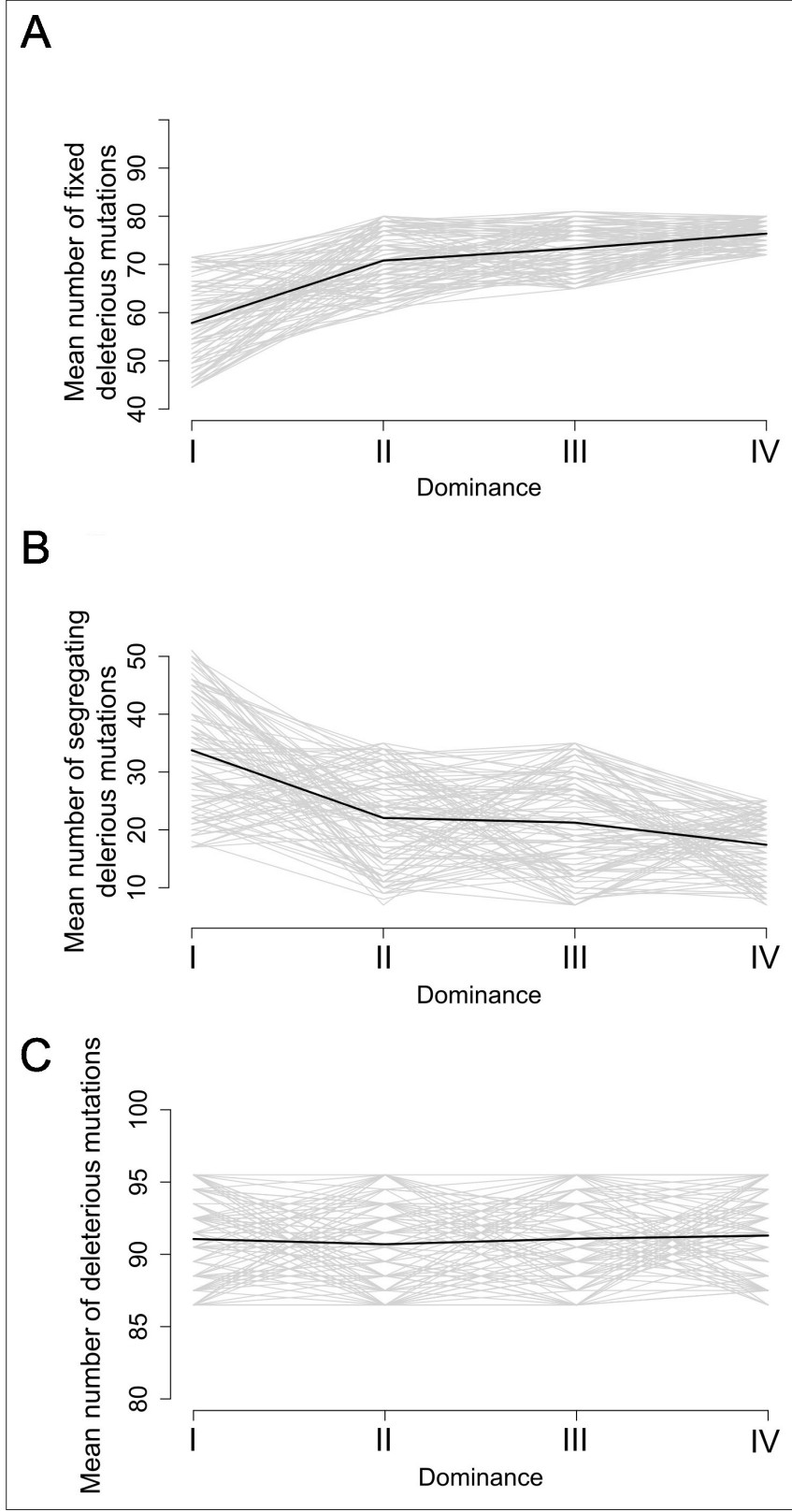

**Figure 5.** Stochastic simulations confirm the contrasted architecture of the load of deleterious mutations linked to dominant vs. recessive S-alleles. Number of fixed (**A**), segregating (**B**), and total (**C**) deleterious mutations linked to *S*-alleles at four different levels of dominance (I<II < III<IV). The means (bold lines) were estimated per *S*-allele dominance classes over 100 replicate simulations after discarding an initial burn-in of 100,000 generations. h = 0. s = 0.01.

## Discussion

### The genetic load linked to the *S*-locus is detectable and manifested on different phenotypes

Our results contribute to a growing body of evidence confirming that the accumulation of deleterious mutations linked to strongly balanced allelic lines can be substantial, and that their effect can be detected at the phenotypic level (*Lane and Lawrence, 1995*; *Stone, 2004*; *Llaurens et al., 2009b*; *Mena-Alí et al., 2009*; *Stift et al., 2013*; *Vieira et al., 2021*). An interesting observation is that the phenotypes on which the load was revealed varied among these studies. Here, the effect of homozygosity at the *S*-locus was apparent on juvenile survival and on the length of the longest flowering stem, but we detected no effect on any other morphological measurements, including leaf and rosette traits. In the same population of *A. halleri*, *Llaurens et al., 2009b* detected an effect on juvenile survival and on leaf size. A study in North American outcrossing populations of *A. lyrata* (*Stift et al., 2013*) detected an effect on juvenile survival, but not on any other traits that they measured. In the horsenettle *Solanum carolinense*, the load was associated with reduced seed viability, flower number, and germination (*Stone, 2004*; *Mena-Alí et al., 2009*). Hence, the most consistent pattern seems to be a decrease of overall juvenile survival, possibly because it is a highly integrative measurement of fitness, whereas other morphological or life history traits can be associated with more specific components of overall fitness.

### A unique genetic load associated with each allele in each population

The model of the sheltered load posits that each *S*-allele should be associated with a specific set of linked mutations (*Llaurens et al., 2009b*). In line with this prediction, the magnitude of the *S*-linked load varied among *S*-alleles as the load linked to some *S*-alleles was phenotypically detectable, while for others it was not. This variation of the genetic load is expected since deleterious mutations associated with the different alleles are likely to hit different linked genes and affect different phenotypic traits with different effects on fitness. Also in line with the model of the sheltered load, our phasing of a large number of variants linked to *S*-haplotypes in several natural populations revealed that the same suite of linked mutations was consistently associated among different copies of a given *S*-allele when sampled from within the same population, in particular for the dominant *S*-allele lineages under more intense balancing selection. As expected for outcrossing populations with short-scale linkage disequilibrium, this association was lost when examining sites at increasing genetic distances from the *S*-locus along the chromosome (see also *Le Veve et al., 2023*). Finally, the association with linked sites was further lost when comparing gene copies of *S*-alleles sampled from different local populations, suggesting that recombination within populations decouples alleles from their linked sites faster than migration can homogenise the genetic composition among these natural populations. We note that the patterns of association and phylogenetic structure differed among populations, possibly due to their contrasted demographic histories. Indeed, the *A. lyrata* populations colonised North America from ancestral European populations about 20–30.000 years ago (*Clauss and Mitchell-Olds, 2006*; *Ross Ibarra et al., 2008*) and are less diverse overall than the *A. halleri* populations we studied, who colonised the north of France during the last century from ancestral German populations (*Pauwels et al., 2005*). The progressive decoupling between alleles and their linked sites leads to the simple prediction that *S*-locus homozygous genotypes formed by crossing individuals carrying identical alleles from distinct populations should not reveal as much load as when they are formed by crossing individuals within populations. Hence, the *S*-locus region could contribute to overall hybrid vigour. Testing this prediction will be an interesting next step.

### Different properties of the linked load according to *S*-allele dominance

The question of whether variations in the intensity of balancing selection, mediated by *S*-allele dominance, could explain variation of the linked load has received conflicting support in the literature. In line with *Stift et al., 2013*, but in contradiction with *Llaurens et al., 2009b*, we observed no overall effect of dominance on the magnitude of the load. Several technical and biological reasons could explain the contrasted results obtained in these different studies. First, phenotypic quantification of the linked load is experimentally demanding, such that these studies relied on the comparison of a limited number of alleles (three *S*-alleles in each of the studies) and therefore each of them had

inherently low power. Second, the experimental procedures to reveal the load varied slightly. *Llaurens et al., 2009b* used $CO_2$ treatment to bypass the SI system and obtain homozygous progenies from crosses that would otherwise have been incompatible, whereas we used the 'natural' masking by dominant *S*-alleles to enable the obtention of recessive homozygous genotypes. Our approach is experimentally simpler and avoids the possible contamination by offspring obtained by selfing, which may confound the effect of the sheltered load with that of genome-wide inbreeding depression (see *Stift et al., 2013* for a detailed discussion of this caveat). Third, a limitation of our approach is that it is restricted to *S*-alleles that are recessive or intermediate along the dominance hierarchy, and is thus not applicable to quantify the load associated with the most dominant *S*-alleles under more intense balancing selection. It is therefore possible that the *S*-alleles we examined did not exhibit sufficiently contrasted levels of dominance, in particular if only the most dominant ones are generating a substantial load, as suggested for fully linked recessive deleterious mutations (*Llaurens et al., 2009b*). In addition, since the homozygous *S*-allele genotypes we created correspond to different gene copies from the population, they may carry distinct sets of linked variants, especially for the more recessive *S*-alleles. The variation we observed in the phenotypic magnitude of the load among families confirms that linked deleterious variants are unlikely to be fixed within all allele lineages. Finally, we note that our genomic analysis of the genetic load shows that the dominant allele Ah15 previously associated with reduced fitness in homozygotes (*Llaurens et al., 2009b*) is indeed unusual in terms of the number of mutations it carries. In fact, it is one of the most 'loaded' alleles among all the dominant *S*-alleles present in this population, possibly explaining why *Llaurens et al., 2009b* observed a significant effect despite the inherently limited experimental power of their analysis.

A possible caveat of the population genetics approach we used is that simply counting up the number of putatively deleterious linked mutations is a very crude estimate of the genetic load. We note that our conclusions are robust to differences in the way we define deleterious mutations: as variants at zerofold degenerate sites, at non-synonymous sites, or using methods to quantify the severity of mutations such as SNPeff of SIFT4G. An obvious limitation is that none of these approaches allow for evaluation of recessivity – a concept critical to ideas concerning the sheltered load. Our stochastic simulations could be improved in several ways. First, we examined the accumulation of linked deleterious mutations that were assumed to be fully recessive. This choice was guided by the observation by *Llaurens et al., 2009b* that fully recessive mutations accumulate more substantially, but it remains possible that the dynamics of deleterious mutations that are only partially recessive may involve a complex interaction with dominance of the S-alleles to which they are linked. Second, allowing for partial recombination between S-alleles and their linked deleterious mutations in the simulations would also be necessary to predict the length of the chromosomal haplotypes associated with dominant vs. recessive *S*-alleles. In spite of these limitations, our stochastic simulations and genomic analyses concur to the conclusion that the variation of the intensity of balancing selection among *S*-alleles affect the genetic architecture of the linked load: a larger proportion of putatively deleterious mutations are fixed among gene copies of the dominant compared to the recessive *S*-alleles, while gene copies of the recessive *S*-alleles tend to accumulate more segregating deleterious variation. While these two processes eventually compensate one another, they may have distinct consequences for the evolution of S-alleles. *Uyenoyama, 2003* showed that the existence of a sheltered load should influence the evolutionary dynamics of new *S*-alleles through self-compatible intermediates. Specifically, antagonistic interactions are expected between ancestral and derived functional specificities because they would initially share their linked deleterious mutations, slowing down the establishment of new *S*-alleles. Our observation that partially different sets of linked mutations are associated with *S*-alleles from the different populations raises the question of whether the (short) time scale at which recombination decouples *S*-alleles from their sets of linked mutation is sufficiently fast to impede such antagonistic interactions to take place. In other words, the effect of the load on the diversification dynamics should be most important if the two mutational steps required for the emergence of new *S*-alleles under this model take place within local populations, rather than involving a metapopulation-scale process. As shown by *Stetsenko et al., 2023*, this is expected to occur under very low dispersal only. In addition, the observation that the architecture of the sheltered load differs between dominant and recessive *S*-alleles suggests that their diversification dynamics may also differ. Specifically, the self-compatible intermediates required for the formation of new *S*-alleles (*Gervais et al., 2011*; *Bergero and Charlesworth, 2009*) are expected to be capable of selfing as well as forming homozygous

genotypes that would otherwise be prevented. While the consequences of selfing may be equivalent for all alleles (because the overall number of mutations to which they are linked are equivalent), the consequences of the formation of homozygotes allowed by the crossing of separate individuals sharing a given S-allele are expected to be more severe for dominant S-alleles. The segregation of distinct deleterious variants linked to different gene copies of recessive S-alleles implies that linked recessive deleterious mutations are likely to remain masked when two distinct gene copies of a given recessive S-allele are brought together. Hence, our results lead to the prediction that in natural populations self-compatible mutants may segregate more readily for the more recessive than for the more dominant S-alleles, and more generally for allelic lineages under lower intensity of balancing selection. Considering that self-compatible mutants are a necessary intermediate stage in the formation of new S-alleles, one may predict that the diversification dynamics should be more efficient for lineages of recessive than dominant S-alleles. This prediction is in line with the observation in *Arabidopsis* that the most dominant S-alleles exhibit the deepest phylogenetic divergence among them (*Durand et al., 2014*). Detailed quantification of the presence of self-compatible variants in natural populations will now be necessary to test this hypothesis. At this stage, however, a proper model of allelic diversification taking into account dominance interactions among S-alleles is still missing.

Variations of the genetic load among balanced allelic lines is a general phenomenon. The classical case of Y or W sex chromosomes are indeed examples where one balanced line accumulates a greater genetic load than the other (X or Z, respectively), eventually leading to substantial genetic degeneration (*Wright et al., 2016*; *Ponnikas et al., 2018*). Another example is the supergene controlling variation in male plumage phenotypes of the ruff, where the genetic load on the derived 'Satellite' haplotype is higher than on the ancestral 'Independent' haplotype (*Lamichhaney et al., 2016*; *Hill et al., 2022*). Similarly, in the butterfly *Heliconius numata*, the inverted haplotypes conferring mimetic wing patterns tend to accumulate a greater load than the non-inverted haplotypes (*Rosser et al., 2022*). Interestingly, in all these cases, the haplotypes with the greatest load also act genetically in a dominant manner, establishing a clear parallel with our observations.

It is clear from our results that S-allele dominance affects the linked load, but in turn the differences in structure of the linked load may affect the conditions under which dominance can evolve. The Brassicaceae S-locus is a unique system, where dominance is controlled by 'dominance modifiers' (*Durand et al., 2014*; *Durand et al., 2020*). The presence of deleterious mutations linked to S-alleles has been shown to affect the evolution of dominance modifiers, favouring evolution towards greater dominance than towards greater recessivity (*Llaurens et al., 2009a*). This asymmetry arises from the fact that S-alleles that become recessive (e.g. following acquisition of a recessivity modifier such as a small RNA target) will start forming homozygous genotypes, leading to expression of their linked load, while S-alleles that become dominant will not. Our observation that many deleterious mutations linked to recessive S-alleles are indeed segregating suggests that expression of the load will be less severe for recessive than for dominant S-alleles, hence decreasing this predicted asymmetry. It will now be essential to modify models for the evolution of dominance to allow for such differential load among S-alleles.

## Materials and methods
### Source plant material

We worked on natural accessions from two closely related species, *A. halleri* and *A. lyrata*, represented by two population samples named Mortagne (50°47'N, 3°47'E, France, n = 60) and Nivelle (50°47'N, 3°47'E, France, n = 61) for *A. halleri*, and three highly outcrossing population samples from the North American Great Lakes, named IND (Indiana Dunes National Lakeshore in Michigan, n = 9), PIN (Pinery Provincial Park in Ontario, n = 11), and TSS (Tobermory Provincial Park in Ontario, n = 8; *Foxe et al., 2010*) for *A. lyrata*. The *A. lyrata* populations colonised North America from ancestral European populations about 20–30,000 years ago (*Clauss and Mitchell-Olds, 2006*; *Ross Ibarra et al., 2008*; *Mattila et al., 2019*) and the *A. halleri* populations are peripheral and likely colonised the north of France during the last century from ancestral German populations (*Pauwels et al., 2005*).

We performed 92, 91, 40, 43, and 21 controlled crosses between randomly chosen individuals within the Nivelle, Mortagne, IND, PIN, and TSS populations, respectively. We successfully obtained seeds from 60, 66, 21, 21, and 10 of these crosses, respectively. Because we were not interested in

estimating population frequencies of *S*-alleles, we instead tried to maximise the number of reconstructed haplotypes and avoid over-representing the most recessive *S*-allele (Ah01) that tends to segregate at very high frequencies in natural populations (*Llaurens et al., 2008*). To do this, we performed PCR with *S*-allele-specific primers (*Llaurens et al., 2008*; *Goubet et al., 2012*) to screen the parents of the crosses and removed from the experiment offspring with two parents carrying allele Ah01. For *A. halleri*, we selected 19 individuals from the Nivelle population and 19 individuals from the Mortagne population based on their genotype at the *S*-locus (*Supplementary file 1g*). We also selected one offspring of 9, 11, 5, 6, and 5 pairs of selected individuals in the Nivelle, Mortagne, IND, PIN, and TSS populations, respectively, for the phasing of *S*-haplotypes (*Supplementary file 1h*). To increase sample size for the phenotypic measurements, we included offspring from five additional crosses from the Nivelle population (*Supplementary file 1h*).

## Library preparation, capture, and sequencing

We used a previously developed sequence capture approach to specifically sequence genomic regions of interest (*Le Veve et al., 2023*). Briefly, indexed genomic libraries were constructed for each individual and libraries were pooled in equimolar proportions. Fragments matching a series of regions of interest (including in particular the 75 kb upstream and downstream of the non-recombining *S*-locus region as well as a series of 100 unlinked 25 kb regions used as genomic controls; *Le Veve et al., 2023*) were then enriched using synthetic 120 bp RNA probes and sequenced by Illumina MiSeq (a total of 159 million paired-end reads).

For six individuals (*Supplementary file 1g and h*), we completed the sequencing with genome-wide resequencing (WGS) in order to distinguish the homozygous and heterozygous genotypes at the *S*-locus based on read depth (*Genete et al., 2020*), which is not possible using data from the capture protocol. The prepared libraries were sequenced by Illumina NovaSeq (2 × 150 pb, paired-end) from the GenoScreen platform (Lille, France).

## Determination of the *S*-locus genotypes and dominance of *S*-alleles

We used a dedicated pipeline for genotyping the *S*-locus based on short reads sequencing (*Genete et al., 2020*) obtained from each individual (*Supplementary file 1g and h*). The level of dominance of *S*-alleles found in our study was determined based on either previous assessment of dominance in *A. lyrata* and *A. halleri* (*Schierup et al., 2001*; *Mable et al., 2003*; *Bechsgaard et al., 2004*; *Llaurens et al., 2008*; *Goubet et al., 2012*) or indirectly inferred based on the observed association between the phylogeny of *S*-alleles and levels of dominance (*Prigoda et al., 2005*).

## Read mapping and variant calling in *A. halleri* and *A. lyrata* populations

Raw reads were mapped on the complete *A. lyrata* reference genome V1.0.23 (*Hu et al., 2011*) using Bowtie2 v2.4.1 (*Langmead and Salzberg, 2012*), as described in *Le Veve et al., 2023*. File formats were then converted to BAM using samtools v1.3.1 (*Li et al., 2009*) and duplicated reads were removed with the MarkDuplicates program of picard-tools v1.119 (http://broadinstitute.github.io/picard). These steps were performed using the custom Python script sequencing_genome_vcf.py available at https://github.com/leveveaudrey/analysis-of-polymorphism-S-locus (copy archived at *Le Veve, 2021a*).

We obtained an average of 620 million properly mapped paired-end 300 bp reads per population sample. For consistency, we conserved only reads which mapped to the *S*-locus flanking or control regions, even for samples sequenced by WGS, using the *targetintercept* option of bedtool v2.25.0 (*Quinlan and Hall, 2010*). We called all SNPs within the chromosomal segment comprising 50 kb upstream from the first base of the gene *Ubox* in 3' and 50 kb downstream from the last base of the gene *ARK3* in 5' of the *S*-locus using the Genome Analysis Toolkit v. 3.8 (GATK; *DePristo et al., 2011*) with the option GVCF and a quality score threshold of 60 using vcftool v0.1.15 (*Danecek et al., 2011*). This region contains 20 annotated protein-coding genes. In this study, we excluded the genes inside the *S*-locus itself (*SCR*, *SRK*). For each sample independently, we computed the distribution of coverage depth across control regions using samtools depth (*Li et al., 2009*). We excluded sites with either less than 15 reads aligned or coverage depth above the 97.5% percentile as the latter are likely to correspond to repeated sequences (e.g. transposable elements or paralogs). Finally, we removed SNPs fixed in each population using the script 1_fix_pos_vcf.py (https://github.com/leveveaudrey/

dominance_and_sheltered_load, copy archived at *Le Veve, 2021b*) thus retaining only nucleotide sites that were variable in the population.

## Quantifying the sheltered load of deleterious mutations

We examined deleterious mutations based on the accumulation of either (1) mutations on zerofold degenerate sites, (2) all non-synonymous mutations, (3) mutations predicted to be deleterious based on the SIFT4G database (*Vaser et al., 2016*), or (4) mutations predicted to be lowly, moderately, and highly deleterious by SNPeff (*Cingolani et al., 2012*). The zerofold and fourfold degenerate sites were identified and extracted from the reference genome and the gene annotation using the script NewAnnotateRef.py (*Williamson et al., 2014*). None of the tools used to predict deleterious mutations are able to determine dominance levels of the mutation. Thus, all the deleterious mutations were considered as recessive. Details of the number of deleterious for each type are presented in *Supplementary file 1f*.

## Phasing *S*-haplotypes

For each of the 9, 11, 5, 6, and 5 trios analysed in the Nivelle, Mortagne, IND, PIN, and TSS populations, respectively, we phased mutations in the flanking regions, resulting in 130 phased haplotypes. Briefly, we used sites that were heterozygous in the offspring to resolve parental haplotypes by assuming no recombination between parent and offspring, thus attributing the allelic state that was shared between a parent and its offspring to their shared *S*-allele, and the allelic state that was not shared to the other (untransmitted) haplotype of the parent. Twelve of the parents had been used in more than one cross, and in these cases we phased their haplotypes only once (*Supplementary file 1h*). We implemented the phasing procedure in the script 3_phase_S_allele.py available at (https://github.com/leveveaudrey/dominance_and_sheltered_load, copy archieved at *Le Veve, 2021b*).

## Study of the structure of *S*-haplotypes

We first developed a new method to evaluate the distortion of the phylogenetic patterns caused by linkage to *S*-alleles. To do this, we used phyml v.3.3 (*Guindon et al., 2010*) to calculate the likelihood of two contrasted topologies of interest: (1) the topology clustering haplotypes by the populations where they came from vs. (2) the topology clustering them by the *S*-allele to which they are linked (*Figure 2A*). We used sliding windows of sequences with 50 SNPs to obtain the variation of the difference in log-likelihood between these two topologies along the chromosome. We then compared these values to their distribution throughout the genome obtained by random draws of sequences with 50 SNPs from the control regions. Second, we visualised the relationships among the phased haplotypes using maximum likelihood phylogenies based on the Tamura-Nei model (*Tamura and Nei, 1993*), with 1000 replicates in MEGA X (*Kumar et al., 2018*). Third, we followed Kamau et al.'s (2007) approach and examined the variation of $F_{ST}$ among populations within each species (Nivelle and Mortagne for *A. halleri* and IND, PIN, and TSS for *A. lyrata*) along the flanking region in non-overlapping windows of 5 kb. We also examined the variation of $F_{ST}$ along the flanking region obtained by grouping haplotypes by their linked *S*-allele rather than by population of origin. Then, we compared these $F_{ST}$ values computed in the *S*-locus flanking regions with their genomic distribution as determined from the 100 control regions. The $F_{ST}$ values were estimated with the DNAsp 6 software (*Rozas et al., 2017*). Fourth, we performed an MCA based on SNPs in the first 5 kb, SNPs between 5 and 25 kb, and SNPs between 25 and 50 kb around the *S*-locus, using the R packages 'ggplot2' (version 3.4.0), 'factoextra' (version 1.0.7), and 'FactoMiner' (version 2.7). We compared the patterns obtained by these MCAs with those obtained from identical numbers of SNP (±1%) from the control regions. Finally, we analysed genetic association in each population independently between each of the locally segregating variants and the *S*-alleles considered as phenotypes using STRAT V1.1 (*Pritchard et al., 2000*) combined with Structure V2.3 (*Pritchard et al., 2010*). We examined the distribution of the top 0.1% most significant associations detected specifically for each *S*-allele in each population.

## Estimation of the number of fixed and segregating deleterious mutations within *S*-allele lineages

For each variable position considered in the phased haplotypes, we estimated the number of mutations on zerofold ($S_{0f}$) and fourfold degenerate sites ($S_{4f}$) compared with the reference genome. We

distinguished SNPs that were fixed from those that were segregating within each of the allelic lines. We used GLM with a Poisson distribution to test whether the number of fixed and segregating mutations was associated with $S$-allele dominance, considering populations as random effects. The dominance of the S-allele was considered a continuous variable. We reiterated the GLM analysis with the number of non-synonymous ($S_{NS}$), synonymous ($S_S$), lowly and moderately deleterious mutations predicted by SNPeff and deleterious mutations predicted by SIFT4G mutations.

## Estimation of the phenotypic impact of homozygosity at the *S*-locus for three *S*-alleles

To determine if the genetic sheltered load putatively linked to the *S*-locus has a detectable phenotypic impact, we performed 45 crosses (*Supplementary file 1a*; *Figure 1—figure supplement 1*) between offspring of the Nivelle individuals that we chose so that they shared one *S*-allele. Based on the dominance hierarchy in pollen (*Durand et al., 2014*; *Supplementary file 1a*), these crosses should correspond to compatible partners. The general principle of the experiment was to take advantage of the dominance hierarchy to mask recessive *S*-alleles and generate full sibs that were either homozygous (because they inherited the *S*-allele that was shared by their two parents) or heterozygous at the *S*-locus, and thus isolate the effect of homozygosity at the *S*-locus. Note that all offspring in our experiments were thus 'naturally' outcrossed, whereas *Llaurens et al., 2009b* based their comparisons on outcrossed progenies obtained by enforced incompatible crosses and *Stift et al., 2013* based their comparisons on enforced selfed progenies. These crosses generated 399 seeds overall, with homozygous genotypes expected for the *S*-alleles Ah01, Ah03, and Ah04 forming the following dominance relationship: Ah01 < Ah03 < Ah04.

Seedlings were grown in a greenhouse between 14.5 and 23.1°C and a photoperiod of 16 hr day/8 hr night. Offspring from the six families were placed on tables, and their position randomised every 3 days. After 3 months of growing, all the germinated plants were vernalised under a temperature between 6 and 8°C and a natural photoperiod for 2 months (January–February). Then, all surviving plants began reproduction in a greenhouse under temperature between 10.6 and 25.3°C and a natural photoperiod. The genotypes at the *S*-locus were determined in surviving plants by a PCR approach using *S*-allele-specific primers for the pistil-expressed *SRK* gene. We assessed the reproductive success of offspring from the different crosses on the basis of 14 phenotypic traits (detailed below) and computed the mean difference for the trait between homozygotes and heterozygotes within each family. We also tested for departures from Mendelian proportions of each *S*-locus genotypic category in the family after the apparition of the first stem. Significant departures were interpreted as reflecting differences in survival between homozygous and heterozygous *S*-locus genotypes. We performed 10,000 replicate simulations of Mendelian segregation based on the *S*-locus genotype of the parents. We used GLM to test whether the phenotypic impact of homozygosity at the *S*-locus increased with dominance of the *S*-alleles, considered as a continuous variable. The models used for GLM depended on the type of trait analysed (Poisson for the counts like the number of leaves, flowers by stems or days; Gaussian for continue traits like the lengths, widths and areas).

We measured the following 14 phenotypic traits: the time (days) to the first leaf measured by visual control every day during 7 weeks after sowing the seeds, the number of leaves, the area of the rosette (cm²), the mean length and width of leaves (cm), the standard deviation of length and width of leaves (cm), and the mean area of leaves (cm²) measured using ImageJ (*Schneider et al., 2012*) based on photographs taken 7 weeks (± 5 days) after the first leaf. At reproduction, we measured the time to the first flower bud for the end of vernalisation (day), scored by visual control every 3 days during 9 weeks, the number of flower buds per flower stem produced during 4 weeks after the appearance of the first bud, the number of flower stems, the length of the highest flower stem produced 4 weeks after the appearance of the first bud (cm), and finally the total duration of buds production (days), scored by visual control every 3 days during 11 weeks after the appearance of the first bud. The last trait we measured was the proportion of homozygotes per family that survived until reproduction assuming Mendelian proportions in the seeds. During the whole experiment, the presence of phytophagous insects, pathogens, and stress markers were scored as binary variables. The presence of phytophagous insects and pathogen attacks was detected by the occurrence of gaps in leaves. Oxidative stress was scored qualitatively based on the occurrence of purple leaves. We also controlled the effect of the family on the phenotypic trait. These effects were controlled by redistributing 10,000 times the values

observed in groups of the same size observed for each effect (e.g. presence or absence of pathogen attack) and comparing the difference for the trait observed with the distribution of the differences obtained in the permutations. We considered the impact of the effect on the trait if the observed difference between groups was higher than the 95% percentile of the distribution obtained randomly (*Supplementary file 1i*). When the test was significant, the effect was implemented as a random effect in the GLM. We used the same method to control for the family effect, which was included as a random effect in GLM if necessary (*Supplementary file 1j*). All data analyses were done in R ver. 3.1.2 (*R Development Core Team, 2021*, *Figure 1—figure supplement 1*).

## Simulations

Finally, we refined the model of *Llaurens et al., 2009b*, in several ways. We simulated a panmictic population of $N$ diploid individuals with non-overlapping generations. Each individual was defined by its genotype in a non-recombining genomic region. This region contains the $S$- locus and a D-locus where deleterious mutations accumulated. For the $S$-locus, we used a simple model of sporophytic SI, with 4 dominance classes, as observed in *A. halleri* (*Genete et al., 2020*; only three classes were considered before in *Llaurens et al., 2009b*), and 14 $S$-alleles (eight alleles in class IV, three in class III, two in class II, and one in class I). This distribution mirrors that of the Nivelle population (*Supplementary file 1g*), with the exception that a class II allele has been added because its presence has been reported in previous studies (*Llaurens et al., 2008*). Alleles within classes were assumed to be codominant with each other and dominant over all alleles of the more recessive classes, with the following linear hierarchy between classes: class I < class II < class III < class IV. We also assumed that no new $S$-allele could appear by mutation during the simulations. The population size was 10,000 diploid individuals, so as to be large enough to avoid $S$-allele loss by drift during the simulations (previously it was 1000). The 'D locus' comprised 100 fully linked biallelic positions (versus a single one in *Llaurens et al., 2009b*). Fully recessive deleterious mutations were recurrently introduced (at a rate $10^{-4}$), and reverse mutations were possible (at a rate $10^{-5}$). We ignored partially recessive deleterious mutations because these mutations were predicted to be effectively eliminated by natural selection in *Llaurens et al., 2009b*. The survival probability $p$ of a zygote depended on its genotype at the D locus: $p = (1 - s)^n$, with $s$ the selection coefficient and $n$ the number of positions homozygous for the mutated allele. We explored different values of the selection coefficient (0.1, 0.05, 0.03, 0.01, and 0.005). Under strong selection ($s = 0.1$, 0.05, and 0.03), the combined effect of multiple mutations led to low-fitness individuals, eventually causing population extinction. Under weak selection, ($s = 0.005$), we observed near fixation of the deleterious mutations under the influence of asymmetrical mutation. Hence, we focused on the intermediate value of the selection coefficient ($s = 0.01$), where deleterious mutations segregated stably in the simulations.

We first ran simulations without deleterious mutations until a deterministic equilibrium for $S$-allele frequencies was reached, which was considered to be attained when allelic frequencies changed by less than $10^{-3}$ between generations. Recessive deleterious mutations were then allowed to accumulate at the positions within the D locus. Each simulation was performed with 100 independent replicates of 100,000 generations, and the frequency of the deleterious alleles was recorded every 1000 generations. At the end of the simulation runs, we estimated the number of deleterious mutations found in each haplotype associated with each $S$-allele to determine the expected patterns of association between the sheltered load and dominance at the $S$-locus.

The code of the program of simulations developed in *Llaurens et al., 2009b* and used in our study is available in GitHub (https://github.com/leveveaudrey/model_ssi_Llaurens, copy archived at *Le Veve, 2024*).

## Acknowledgements

This work was funded by the European Research Council (NOVEL project, grant #648321) and ANR TE-MoMa (grant ANR-18-CE02-0020-01). AL's PhD thesis was funded by the ERC and the University of Lille. The authors thank Barbara Mable for sharing seeds of *A. lyrata* and Camille Roux for discussions. This work was performed using infrastructure and technical support of the Plateforme Serre, cultures et terrains expérimentaux - Université de Lille for the greenhouse/field facilities. The authors thank the UMR 8199 LIGAN-MP Genomics platform (Lille, France) which belongs to the 'Federation de Recherche' 3508 Labex EGID (European Genomics Institute for Diabetes; ANR-10-LABX-46) and

was supported by the ANR Equipex 2010 session (ANR-10-EQPX-07-01; 'LIGAN-MP'). The LIGAN-MP Genomics platform (Lille, France) is also supported by the FEDER and the Region des Hauts-de-France. The authors thank the GenoScreen platform (Lille, France).

## Additional information

### Competing interests
Vincent Castric: Reviewing editor, eLife. The other authors declare that no competing interests exist.

### Funding

| Funder | Grant reference number | Author |
|---|---|---|
| European Research Council | #648321 | Vincent Castric |
| Agence Nationale de la Recherche | ANR-18-CE02-0020-01 | Vincent Castric |

The funders had no role in study design, data collection and interpretation, or the decision to submit the work for publication.

### Author contributions
Audrey Le Veve, Conceptualization, Data curation, Formal analysis, Investigation, Methodology, Writing – original draft, Writing – review and editing; Mathieu Genete, Data curation, Software, Methodology; Christelle Lepers-Blassiau, Chloé Ponitzki, Data curation, Methodology; Céline Poux, Validation, Writing – original draft; Xavier Vekemans, Resources, Supervision, Writing – original draft, Project administration, Writing – review and editing; Eleonore Durand, Supervision, Writing – original draft; Vincent Castric, Supervision, Funding acquisition, Investigation, Writing – original draft, Project administration, Writing – review and editing

### Author ORCIDs
Audrey Le Veve  https://orcid.org/0000-0001-5895-370X
Vincent Castric  https://orcid.org/0000-0002-4461-4915

Joint Public Review: https://doi.org/10.7554/eLife.94972.3.sa1
Author response https://doi.org/10.7554/eLife.94972.3.sa2

## Additional files

### Supplementary files
• Supplementary file 1. Protocol used. (a) Crosses performed to obtain homozygotes for three S-alleles. (b) Trait variation in S-locus homozygous individuals. (c) Trait variation in homozygous at the S-locus for the S-alleles Ah01 and Ah04 between families. (d) Effect of dominance on variation of phenotypic traits in S-locus homozygous individuals. (e) Number of phased haplotypes linked to S-alleles in each sample. (f) Effect of dominance on the accumulation of genetic load in S-flanking regions. (g) S-locus genotypes of individuals sequenced using the capture protocol. (h) S-locus genotypes of the offspring selected for haplotype phasing and the crosses for the study of phenotypic traits. (i) Effect of phytopathogen, phytophagous attacks and oxidative stress on the phenotypic traits. (j) Difference on the phenotypic traits variation between the two families for each allele tested.

• MDAR checklist

### Data availability
All sequence data are available in the NCBI Short Read Archive (SRA; https://www.ncbi.nlm.nih.gov/sra) with accession codes: PRJNA744343, PRJNA755829. All scripts developed are available in Github (https://github.com/leveveaudrey/dominance_and_sheltered_load, copy archieved at *Le*

*Veve, 2021b*), (https://github.com/leveveaudrey/analysis-of-polymorphism-S-locus, copy archived at *Le Veve, 2021a*).The code of the program of simulations developed in *Llaurens et al., 2009a* and used in our study is available in Github (https://github.com/leveveaudrey/model_ssi_Llaurens, copy archived at *Le Veve, 2024*).

The following dataset was generated:

| Author(s) | Year | Dataset title | Dataset URL | Database and Identifier |
|---|---|---|---|---|
| Audrey LV, Mathieu G, Christelle L-B, Chloé P, Céline P, Xavier V, Eleonore D, Vincent C | 2024 | impact of dominance on genetic load linked at the S locus | https://www.ncbi.nlm.nih.gov/bioproject/?term=PRJNA755829 | NCBI BioProject, PRJNA755829 |

The following previously published dataset was used:

| Author(s) | Year | Dataset title | Dataset URL | Database and Identifier |
|---|---|---|---|---|
| Le Veve A, Burghgraeve N, Genete M, Lepers-Blassiau C, Takou M, De Meaux J, Mable BK, Durand E, Vekemans X, Castric V | 2023 | study of genetic diversity at the S locus | https://www.ncbi.nlm.nih.gov/bioproject/PRJNA744343 | NCBI BioProject, PRJNA744343 |

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
