## [Editor Report · eLife assessment]

This study presents **valuable** empirical work and simulations that are relevant for the evolution of genetic load linked to self-incompatibility alleles in two *Arabidopsis* species. The evidence supporting the findings is **solid**, although it remains to be seen how generalisable the conclusions are beyond the specific system investigated here, not least because the statistical significance varied between the two species. The work will be of relevance to geneticists interested in the evolution of allelic diversity in similar systems.

---

## [Referee Report · Joint Public Review]

An outside expert evaluated your responses to the original reviewers and offered the following comments:

The main criticism was whether deleterious variants were appropriately classified in the work. The authors use two different methods to characterize the effect of alleles to satisfy these comments. The result is somewhat complex. The authors do replicate the effect of dominance on fixation and segregation of deleterious alleles by classifying polymorphisms as synonymous or synonymous with SNPeff. This is not entirely surprising as it is approximately equivalent to classifying based on fold degeneracy (but it includes sites that have other than 0 or 4 fold degeneracy). However, the authors do not mention in the text that their observation of increased segregating deleterious mutations in recessive alleles was only statistically significant in A. halleri (for both analyses). Using SIFT, the authors only find an effect of dominance in A. lyrata. So in reality, while the trends are the same across the analyses, the statistical significance of the effects of dominance was not consistent.

Reviewer 2 had several more detailed criticisms of the manuscript. The first was that the authors should explore the dominance of linked deleterious mutations themselves. I agree that this would be interesting, but it is very difficult to accomplish, and I agree with the author's reluctance to do much more here. The reviewer also criticized the authors simulation approach. The authors provided their simulation script as requested, but declined to do additional simulations under varied selection coefficients. I felt this was a minimally adequate response to the reviewers concerns, but the authors could have reasonably conducted a few additional simulations under varied selection coefficients.

I think that the scope of the findings described in the assessment was reasonable. This is interesting work, but despite the author's arguments, the system is somewhat unique if for no other reason than that balancing selection at S-loci is uniquely strong

---

## [Author Response]

The following is the authors’ response to the original reviews.

**Reviewer #1 (Public Review):**
Summary:The paper combines phenotypic and genomic analyses of the "sheltered load" (i.e. the accumulation of deleterious mutations linked to S-loci that are hidden from selection in the homozygous state) in Arabidopsis. The authors compare results to previous theoretical predictions concerning the extent of the load in dominant vs recessive S-alleles, and further develop exciting theory to reconcile differences between previous theory and observed results.Strengths:This is a very nice combination of theory and data to address a classical question in the field.

We thank the reviewer for this positive feedback.

Weaknesses:The "genetic load" is a poorly defined concept in general, and its quantification via the number of putatively deleterious mutations is quite difficult. Furthermore counting up the number of derived mutations at fully constrained nucleotides may not be a great estimate of the load, and certainly does not allow for evaluation of recessivity -- a concept critical to ideas concerning the sheltered load. Alternative approaches - including estimating the severity of mutations - could be helpful as well. This imperfection in available approaches to test theory must be acknowledged more strongly by the authors.

As suggested by the reviewer, we implemented alternative approaches to estimate the severity of deleterious mutations and now report the results of SNPeff and

SIFT4G analyses in Table S6. The results we obtained with these other metrics were overall very similar to those based on our previous counting of mutations at 0-fold and 4-fold degenerate sites. More generally, we tried to improve the presentation of our strategy to estimate the genetic load clarified in lines 262-268, 271, 292-295, 297. In particular, we made it clear that our population genetic analysis cannot assess the recessivity of the observed mutations (lines 428-434).

**Reviewer #2 (Public Review):**
Summary:This study looks into the complex dominance patterns of S-allele incompatibilities in Brassicaceae, through which it attempts to learn more about the sheltering of deleterious load. I found several weak points in the analyses that diminished my excitement about the results. In particular, the way in which deleterious mutations were classified lacked the ability to distinguish the severity of the mutations and thus their expected associated dominance.

First, we would like to clarify that our goal with this study is NOT to learn something about dominance of the linked deleterious mutations (we can not). Instead, we compare the accumulation of deleterious mutations linked to dominant vs recessive S-ALLELES, but are agnostic regarding the dominance level of the LINKED mutations themselves. The rationale is that the different intensities of natural selection between dominant vs recessive S-alleles provide a powerful way to examine the process by which deleterious mutations are sheltered in general. We further clarified this aspect on lines 70-73 and 399-401.

Second, as mentioned above in response to Reviewer 1, we complemented the analysis by predicting the severity of the deleterious mutations by SIFT4G and SNPeff. The results were largely consistent, with the exception that the number of sites included in SIFT4G was low, such that the statistical power was reduced (lines 296-300).

Furthermore, the simulation approach could have provided this exact sort of insight but was not designed to do so, making this comparison to the empirical data also less than exciting for me.

As explained above, studying dominance of the linked mutations we observed is an interesting research question (albeit a difficult one), but it was not our goal here. Instead, our study was designed as an empirical test of the predictions presented in Llaurens et al (2009), and we re-analysed some aspects of the model outcome to illustrate our points.

We now better explain that we based our choice of parameters on the fact that in the theoretical study by Llaurens et al (2009), recessive deleterious mutations are predicted to accumulate in a much more straightforward manner (line 316-318).

We now dedicate a paragraph of the discussion to explain how our stochastic simulations could be improved, and acknowledge that a full exploration of the interaction between dominance of the S-alleles and dominance of the linked deleterious mutations would be an interesting follow-up - albeit beyond the scope of our study (line 437-441).

Major and minor comments:I think the introduction (or somewhere before we dive into it in the results) of the dominance hierarchy for the S-alleles needs a more in-depth explanation. Not being familiar with this beforehand really made this paper inaccessible to me until I then went to find out more before continuing. I would expect this paper to be broad enough that self-contained information makes it accessible to all readers. For example, lines 110-115 could be in the Introduction.

We thank the reviewer for this useful remark. We now give a more comprehensive description of the dominance hierarchy and introduce the classes of dominance in A. lyrata already in the introduction, on lines 64-70.

Along with my above comment, perhaps it is not my place to comment, but I find the paper not of a broad enough scope to be of interest to a broad readership. This S-allele dominance system is more than simple balancing selection, it is a very complex and specific form of dominance between several haplotypes, and the mechanism of dominance does not seem to be genetic. I am not sure that it thus extrapolates to broad comments on general dominance and balancing selection, e.g. it would not be the same as considering inversions and this form of balancing selection where we also expect recessive deleterious mutations to accumulate.

We disagree with these interpretations by the reviewer, for two reasons:

First, the mechanism of dominance is actually entirely genetic. In fact, we uncovered some years ago that it is based on the molecular interaction between small non-coding RNAs from dominant alleles and their target sites on recessive alleles (Durand et al. Science 2014, see lines 68-70). If there is something specific with this system, it is that the dominance phenomenon is better understood at the mechanistic level than in most other cases, but the resulting phenomenon in itself (a dominance hierarchy) is rather common.

Second, the kind of variation in the intensity of linked selection created by this mechanism is actually a general phenomenon, so our results have broad relevance beyond our particular study system. We modified the introduction to explain this point

more clearly, highlighting in particular the fact that the situation we study closely resembles the case of sex chromosomes, where X (or Z) chromosomes are genetically recessive and Y (or W) chromosomes are genetically dominant. We cite this example in lines 83-87 of the introduction and also several well-studied other examples on lines 480-489 of the discussion.

It would have been particularly interesting, or a nice addition, to see deleterious mutations classed by something like SNPeff or GERP where you can have different classes of moderate to severe deleterious variants, which we would expect also to be more recessive the more deleterious they are. In line with my next comment on the simulations, I think relative differences between mutations expected to be more or less dominant may be even more insightful into the process of sheltering which may or may not be going on here.

We agree with the reviewer, and as detailed above we have now integrated such analyses with SNPeff and SIFT4G (Table S6). These new results reinforce our conclusion that while S-allele dominance influences the fixation of deleterious mutations, it has no effect on their total number. See lines 270-272 and 296-300.

In the simulations, h=0 and s=0.01 (as in Figure 5) for all deleterious mutations seems overly simplistic, and at the convenient end for realistic dominance. I think besides recessive lethals which we expect to be close to h=0 would have a much larger selection coefficient, and other deleterious mutations would only be partially recessive at such an s value. I expect this would change some of the simulation results seen, though to what degree I am not certain. It would be nice to at least check the same exact results for h=0.3 or 0.2 (or additionally also for recessive lethals, e.g. h=0 and s=-0.9). I would also disagree with the statement in line 677, many studies have shown, particularly those on balancing selection, that partially recessive deleterious mutations are not eliminated by natural selection and do play a role in population genetic dynamics. I am also not surprised that extinction was found for higher s values when the mutation rate for such mutations was very high and the distribution of s values was constant. An influx of such highly deleterious mutations is unlikely to ever let a population survive, yet that does NOT mean that in nature, the rare influx of such mutations does lead to them being sheltered. I find overall that the simulation results contribute very little, to none, to this paper, as without something more realistic, like a simultaneous distribution of s and h values, you cannot say which, if any class of these mutations are the ones expected to accumulate because of S-allele dominance.

We understand that the previous version of our manuscript was confusing between dominance of the S-alleles and dominance of the linked deleterious mutations. We clarified that our study focuses on the effect of the former only (lines 99, 263-264 and 581-583).

We agree that a complete exploration of the interaction between dominance of the S-alleles and dominance of the linked mutations being sheltered would have been an asset, but as explained above this is not the focus of our study. The previous work by Llaurens et al (2009) has already established that deleterious mutations can fix within S-allele lineages, especially when linked to dominant S-alleles, and when the number of S-alleles is large. Under the conditions they examined, deleterious mutations were much more strongly eliminated if not fully recessive (h=0 vs h=0.2), so for the present study we decided to simulate fully recessive mutations only. We now formally acknowledge the possibility that some complex interaction may take place between dominance of the S-alleles and dominance of the linked deleterious mutations (lines 440-442). However, as explained above we feel that fully exploring this complex interaction would require a detailed investigation, which is clearly beyond the scope of the present study.

Rather they only show the disappointing or less exciting result that fully recessive, weakly deleterious mutations (which I again think do not even exist in nature as I said above) have minor, to no effect across the classes of S-allele dominance. They provide no insight into whether any type of recessive deleterious mutation can accumulate under the S-allele dominance hierarchy, and that is the interesting question at hand. I would either remove these simulations or redo them in another approach. The authors never mention what simulation approach was used, so I can only assume this is custom, in-house code. Yet I do not find that code provided on the github page. I do not know if the lack of a distribution for h and s values is then a choice or a programming limitation, but I see it as one that should be overcome if these simulations are meant to be meaningful to the results of the study.

The code we used (in C) was adapted from the previous study by Llaurens et al. (2009), which at the time was not deposited in a data repertory, unfortunately. With the agreement of the authors of that study, this code is now available on Github:

(https://github.com/leveveaudrey/model_ssi_Llaurens; line 723).

It is correct that our simulations were not aimed at determining whether “any type of recessive deleterious mutation can accumulate”, but we strongly believe that they help interpreting the observations made in the genomic data.

**Recommendations for the authors:**
Notes from the editor:I found Table 1 confusing, with column headings of observed proportion but perhaps numbers reflecting counts.

Thank you for pointing out this confusion. There was indeed an error in the last column, which we have now corrected.

I found Figure 2 a bit hard to parse, with the vertical lines being unclear and the x-axis ticks of insufficient resolution to evaluate the physical extent of the signals.

We increased the size of the label on the x-axis and detailed it on the Figure 2, which is now hopefully more clear. Moreover, we increase the size of the vertical lines.

Finally, I wonder, given the rapid decay of signal in lyrata, whether 25kb is the right choice for evaluating load and whether the pattern may look different on a smaller scale.

It is true that the signal decays rapidly in A. lyrata, as can be seen in the haplotype structure analysis and in line with our previous analysis of the same populations Le Veve et al (MBE 2023; in this study we explored the effect of the choice of the size of the chromosomal region analyzed; lines 266-269). However, for the sake of comparison, we prefer to stick to the same window size. The fact that we still see an effect of dominance in spite of the lower statistical power associated with the more rapid decay (because a smaller number of genes is expected to be impacted) actually reinforces our conclusions.

**Reviewer #1 (Recommendations For The Authors):**
I have a few additional suggestions to improve the manuscript.(1) How does the load linked to the S-locus compare to that observed in other genomic regions? It would be useful to provide a comparison of the results quantified in Figures three and four to comparable genomic regions unlinked to the S-locus. How severe is the linked load?

This comparison to the genomic background was actually the core of our previous study (Le Veve et al MBE 2023), which was based on the same populations. This analysis revealed that polymorphism of the 0-fold degenerate sites was more than twice higher in the 25kb immediately flanking the S-locus than in a series of 100 unlinked control regions. Here, the main focus of the present study is on the effect of linkage to particular S-alleles (which was not possible previously because haplotypes had to be phased).

(2) Details of the GLM for data underlying Figures 3 and 4 are somewhat unclear. Is the key explanatory variable (Dominance) treated as continuous? Categorical? Ordinal etc…

Dominance is considered as a continuous variable. We specify this in line 162 of the results, in the legends of Figures 3 and 4, in the Material and Method (lines 627 and 660) and in the legend of Table S4.

(3) I had some trouble understanding the two different p-values in columns five and six of table one. Please provide more detail.

We understand that the two p-values in Table 1 were confusing. The first was related to the binomial test and the second to the permutation test. To be consistent with the rest of the manuscript, we conserved only the p-value of the permutation test.

(4) As mentioned in the "weaknesses" above, the authors should be more clear about what they are quantifying. They are explicitly counting the number of variants at 0-fold degenerate sites as a proxy for the genetic load. How good this proxy is is unclear. The most egregious misstatement here was on line 314 in which they make reference to the "total load." However, this limitation should be acknowledged throughout the manuscript and deserves more attention in the methods and discussion.

As mentioned above, we now integrate additional methods to define and quantify the load (SIFT4G and SNPeff), which reinforced our previous conclusions (lines 271-272, 297-302).

We clarified our wording and replaced the mention of “total load” by “mean number of linked deleterious mutations per copy of S-allele” (line 324-325). In the discussion we tried to better explain the limitations of approaches to estimate the genetic load (line 431-437).

**Reviewer #2 (Recommendations For The Authors):**
Line 60, it should be specified that this is only for recessive deleterious mutations.Non-recessive deleterious mutations would certainly not be expected to accumulate.

As explained in details above, the question of whether and how non-recessive deleterious mutations can accumulate when linked to the S-locus is difficult and would in itself deserve a full treatment, which is clearly beyond the scope of the present study. We clarified this point on line 56.